# CryoEM structure of the human SLC4A4 sodium-coupled acid-base transporter NBCe1

Kevin W. Huynh[1,2], Jiansen Jiang[2,3], Natalia Abuladze[1], Kirill Tsirulnikov[1], Liyo Kao[1], Xuesi Shao[4],
Debra Newman[1], Rustam Azimov[1], Alexander Pushkin[1], Z. Hong Zhou[2,3] & Ira Kurtz[1,5]

Na$^+$-coupled acid–base transporters play essential roles in human biology. Their dysfunction has been linked to cancer, heart, and brain disease. High-resolution structures of mammalian Na$^+$-coupled acid–base transporters are not available. The sodium-bicarbonate cotransporter NBCe1 functions in multiple organs and its mutations cause blindness, abnormal growth and blood chemistry, migraines, and impaired cognitive function. Here, we have determined the structure of the membrane domain dimer of human NBCe1 at 3.9 Å resolution by cryo electron microscopy. Our atomic model and functional mutagenesis revealed the ion accessibility pathway and the ion coordination site, the latter containing residues involved in human disease-causing mutations. We identified a small number of residues within the ion coordination site whose modification transformed NBCe1 into an anion exchanger. Our data suggest that symporters and exchangers utilize comparable transport machinery and that subtle differences in their substrate-binding regions have very significant effects on their transport mode.

[1] Department of Medicine, Division of Nephrology, David Geffen School of Medicine, University of California, Los Angeles, CA 90095, USA. [2] California NanoSystems Institute, University of California, Los Angeles, CA 90095, USA. [3] Department of Microbiology, Immunology & Molecular Genetics, University of California, Los Angeles, CA 90095, USA. [4] Department of Neurobiology, University of California, Los Angeles, CA 90095, USA. [5] Brain Research Institute, University of California, Los Angeles, CA 90095, USA. These authors contributed equally: Kevin W. Huynh, Jiansen Jiang, Natalia Abuladze and Kirill Tsirulnikov. Correspondence and requests for materials should be addressed to A.P. (email: apushkin@mednet.ucla.edu) or to Z.H.Z. (email: Hong.Zhou@ucla.edu) or to I.K. (email: ikurtz@mednet.ucla.edu)

In mammals, solute carrier (SLC) transporters within the amino-acid-polyamine-organocation (APC) protein super-family regulate blood pressure, ion and metabolite home-ostasis, acid–base chemistry, and play key roles in the maintenance of cell function and growth[1–14]. The dysregulation of Na$^+$-coupled acid–base SLC transporters in cancer cells has important diagnostic and therapeutic implications[1]. In the brain, neuronal excitability is modulated by these transporters[2], and the electrical and contractile properties of cardiac myocytes[3] are influenced by changes in their transport activity. Of the afore-mentioned transport proteins, the importance of SLC4 bicarbonate transporters in human biology is highlighted by the diseases associated with their functional loss including blindness, short stature, abnormal cognitive function, cerebral calcification, metabolic acidosis, anemia and hearing abnormalities[4,14]. Among the SLC4 transporters, Na$^+$-dependent members function as either symporters or exchangers that mediate their specific phy-siologic roles in different cell types by thermodynamically cou-pling the transport of Na$^+$ to base (HCO$_3^-$ or CO$_3^{2-}$) with or without simultaneous Cl$^-$ transport. Despite the importance of these transporters in human biology and disease, high-resolution structures of Na$^+$-coupled SLC4 and other eukaryotic Na$^+$-cou-pled acid–base transporters do not exist. To further improve our general understanding of these transporters, we investigated the structural and functional properties of human NBCe1, a Na$^+$-coupled SLC4 symporter, using cryo electron microscopy (cryoEM) and functional mutagenesis studies. The 3.9 Å structure of NBCe1 provides a common framework for the Na$^+$-coupled SLC4 transporters that underlies their functional properties.

## Results

**NBCe1 structure.** Full-length human NBCe1 was expressed in mammalian HEK-293 cells to preserve the proper folding and function of the transporter[15]. NBCe1 was solubilized in 1% Triton X-100, purified and exchanged into PMAL-C8 amphipol resulting in well-dispersed particles (Supplementary Fig. 1a–c). Movie recording with electron-counting detection technology[16] and frame alignment with dose-weighting[17], together allowed for the visualization and subsequent two-dimensional (2D) image clas-sification (Fig. 1a insets and Supplementary Fig. 1c) of this rela-tively small (~130 kDa) membrane protein. The 2D class averages revealed various views of particles with high-resolution structural features (helices) enclosed within a smooth, low-resolution (amphipol) envelope (Fig. 1a insets and Supplementary Fig. 1c). By combining ~185,000 best particles, we obtained a cryoEM reconstruction at near-atomic (3.9 Å) resolution (Fig. 1a, Table 1 and Supplementary Fig. 1c–f). The reconstruction contains two NBCe1 monomers with identical structures and its side view resembles the iconic double-headed eagle with its body, wing, head and foot corresponding to the gate domain, the core domain, extracellular loop 3 (EL3) domain and the cytoplasmic region of each monomer, respectively (Fig. 1a, Supplementary Movie 1). Thus, NBCe1 in solution exists as a homodimer, consistent with our biochemical analysis (Supplementary Fig. 1a). The three-dimensional (3D) structure shows a belt of amphipol density surrounding the NBCe1 homodimer (Fig. 1a and Sup-plementary Movie 1). Bulky amino-acid side chains are resolved in the transmembrane helices (TMs) of each monomer (Supple-mentary Fig. 2 and Supplementary Movie 2), whereas the amphiphilic densities are featureless, suggesting that the inter-action between amphipol and NBCe1 is non-specific. Using the bulky side chains visible in our cryoEM density map as landmarks and incorporating results from sequence-based structure predic-tion (see Methods), we built an atomic model of the NBCe1 homodimer (Fig. 1b and Supplementary Movie 1). Each NBCe1

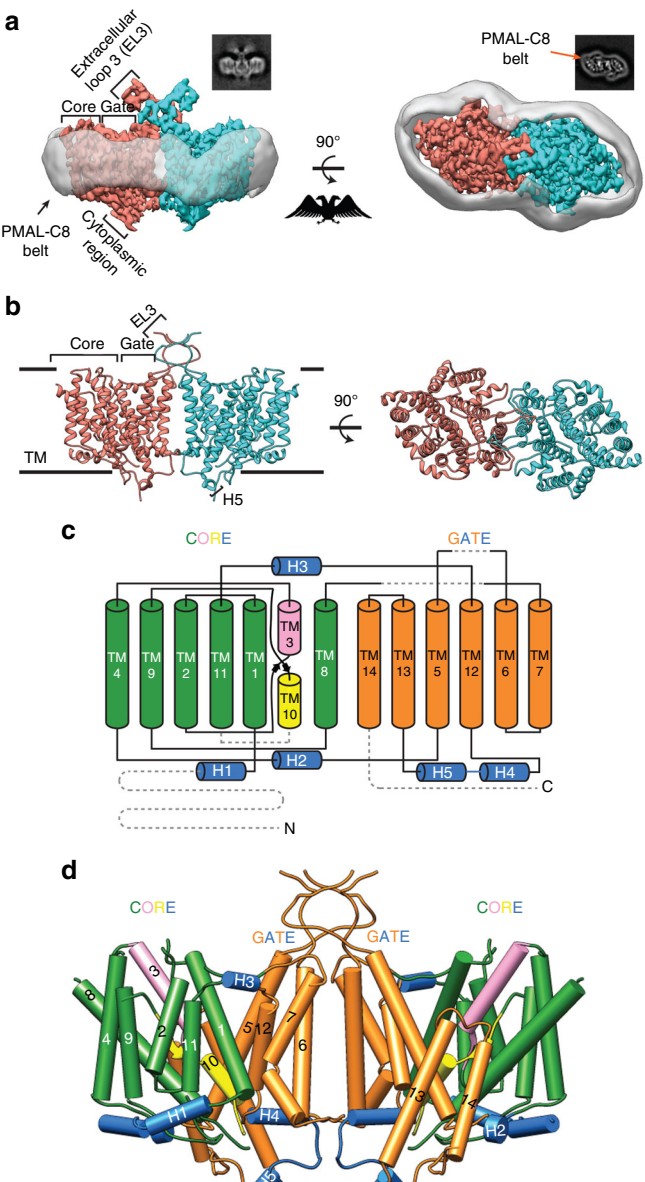

**Fig. 1** CryoEM reconstruction and model of human NBCe1. **a** In all, 3.9-Å cryoEM densities of the NBCe1 dimer showing side (left) and top (right) views. Individual monomers are colored differently. The transparent gray density represents the amphipol belt. The corresponding 2D class averages are shown in the insets. **b** Ribbon model (side and top views) of the NBCe1 dimer. **c** Topology of NBCe1. TMs 1, 2, 4, 8, 9, 11 (green cylinders) together with TM3 (pink) and TM10 (yellow) belong to the core domain. TMs 5, 6, 7, 12, 13, and 14 (orange cylinders) form the gate domain. Helices 1–5 (H1–5) are marked blue. Dotted gray lines represent unmodeled regions. **d** Three-dimensional (3D) model of the NBCe1 dimer. TMs 1–14 and helices H1–5 are shown as cylinders

monomer consists of 14 TMs (divided into two structurally related inverted repeats: TMs 1–7 and TMs 8–14), with four amphipathic helices (H1-4), a short (single turn) cytoplasmic helix (H5), and the loops connecting all these helices (Fig. 1c, d, Supplementary Fig. 3). Densities for the cytoplasmic N-terminus (Met1-Gly400), the distal portion of EL3 (Cys583-Leu637), EL4 (Ser713-Trp734), intracellular loop 5 (IL5; Glu814-Lys825) and the cytoplasmic C-terminus (Asp966-Cys1035) were less well-resolved due to the dynamic nature of these regions and were not modeled (Fig. 1b–d).

**Table 1 Cryo-EM data collection, refinement, and validation statistics**

| Data collection and processing | |
| --- | --- |
| Magnification | 36,764 |
| Voltage (kV) | 300 |
| Electron exposure (e–/Å$^2$) | 24 |
| Defocus range (μm) | −1.4 to −3.2 |
| Pixel size (Å) | 0.68 (Super-resolution mode) |
| Symmetry imposed | C2 |
| Initial particle images (no.) | 1,729,419 |
| Final particle images (no.) | 184,561 |
| Map resolution (Å) | 3.9 |
| FSC threshold | 0.143 |
| Map resolution range (Å) | 3-5 |
| *Refinement* | |
| Model resolution cutoff (Å) | 3.9 |
| FSC threshold | 0.143 |
| Map sharpening B-factor (Å$^2$) | −240 |
| Model composition | |
| Non-hydrogen atoms | 7344 |
| Protein residues | 948 (474 per protomer) |
| Ligands | 0 |
| R.m.s. deviations | |
| Bond lengths (Å) | 0.009 |
| Bond angles (°) | 1.288 |
| *Validation* | |
| MolProbity score | 2.11 |
| Clashscore | 9.23 |
| Poor rotamers (%) | 0 |
| Ramachandran plot | |
| Favored (%) | 86.99 |
| Allowed (%) | 13.01 |
| Disallowed (%) | 0 |

Our atomic model of the NBCe1 monomer is arranged into gate and core domains (Fig. 1c, d). The boundaries between the gate and core domains are well defined and clearly visible in the top views of both the 3D structure (Supplementary Fig. 4) and the 2D class averages (Fig. 1a and Supplementary Fig. 1c). The gate domain is composed of six TMs (TMs 5–7 and 12–14), amphipathic helix H4 and short cytoplasmic helix H5 (Fig. 1c, d). The cytoplasmic region of the cryoEM densities consists of the H4-loop-H5 that bridges TM12 to TM13 of the gate domain (Fig. 1d). The core domain consists of eight TMs (TMs 1–4 and 8–11) and amphipathic helices H1-3 (Fig. 1c, d). Helix H1 is located on the cytoplasmic side and is parallel to the plasma membrane before connecting to TM1. Helices H2 and H3 link the core domain to the gate domain at the cytoplasmic and extracellular sides, respectively. TMs 2, 4, 9, and 11 are nearly vertically oriented, and appear as dots next to the amphiphilic belt in the top view of the 2D class averages (Fig. 1a).

**Ion accessibility pathway and coordination site**. The region corresponding to the middle of the transporter within each monomer formed by short TM3 and TM10 and the antiparallel β-strands preceding these TMs (Fig. 1c, d and Supplementary Movie 3) has been proposed to contribute to substrate coordination in other transporters[18–22]. This region of NBCe1 is accessible from the extracellular side (Fig. 2a, b) as predicted by HOLE2[23] representing an outward-open conformation. The ion accessibility pathway predicted by HOLE2 is located between the core and the gate domain and is formed by TMs 1, 3, 5, 8, 10, 12, and a short EL7 connecting TM13 and TM14 (Fig. 2a). The

pathway contains an electropositive opening from the extracellular side (Supplementary Fig. 5a) with an external diameter >12 Å and the most constricted region (~2 Å diameter) formed by TM3, TM10, and the antiparallel β-strands in the middle of each NBCe1 monomer. Furthermore, residues Phe443, Leu446, Asp555, and Lys559, whose substitution significantly alters NBCe1 function, line the ion accessibility pathway (Fig. 2c)[24-26]. We propose that the inhibition of NBCe1 by disulphonic stilbenes such as 4,4'-Diisothiocyano-2,2'-stilbenedisulfonic acid (DIDS) via its interaction with Lys559[27] is mediated by blocking the ion accessibility pathway and/or the conformational changes during the transport cycle (Supplementary Fig. 5b).

We previously hypothesized that the disease-causing T485S and G486R mutations reside within the NBCe1 ion coordination site[28]. Our NBCe1 structural model shows that these residues are located in the region formed by short TMs 3 and 10 with the antiparallel β-strands preceding these TMs, and TM8 outlining the ion accessibility pathway (Fig. 3a). The side chain of Thr485 is facing the pathway (Fig. 3a), and is likely involved in ion coordination (Fig. 3b). The adjacent G486R patient mutation induces a significant decrease in NBCe1 function. The position of Gly486 in our model suggests that the mutation to the large arginine residue blocks access to the ion coordination site. In addition to Thr485, our structure and mutagenesis data suggests that Ser483 (the β-strand preceding TM3), Ser484 (the β-strand preceding TM3), Asp754 (TM8), Thr758 (TM8), and Thr801 (TM10) are also involved in ion coordination (Fig. 3). Asp754 is specific for the Na$^+$-coupled acid–base transporters within the SLC4 family[4,14]. Substitution of Asp754 with glutamate significantly impaired NBCe1 function (Fig. 3b). The Na$^+$-independent members of the SLC4 family mediating anion exchange (AE1-AE3) have a glutamate residue in this position (AE1 shown in Fig. 4a) that has been proposed to be a proton acceptor[29] suggesting that the substitution of glutamate for aspartate favors Na$^+$-coupling in the Na$^+$-dependent SLC4 transporters. Furthermore, the NBCe1 Ile803 (TM10) residue is in close proximity to the ion coordination site and corresponds to the functionally important Arg730 in AE1[30] (Fig. 4b). Substitution of Ile803 to arginine blocked NBCe1 function (Fig. 3b), suggesting that the presence of isoleucine at this location in NBCe1 (and all Na$^+$-coupled SLC4 acid–base transporters) instead of the positively charged arginine in AE1, also favors Na$^+$-coupling.

**Conversion into an anion exchange transport mode**. NBCe1 and AE1[22] have a similar fold yet differ significantly with regards to both their ion transport preferences and mode of transport (exchanger vs. symporter respectively). Based on our NBCe1 structure, functional data, and alignment analysis, we hypothesized that the very different functional properties of NBCe1 and AE1 are mediated by specific stretches of residues within TM8 and TM10, and the antiparallel β-strands preceding TM3 and TM10. We designed a series of NBCe1 constructs (Fig. 4a) with specific combinations of amino acids substituted with corresponding residues in the predicted AE1 ion coordination site (Fig. 4b). As shown in Figs. 4c–m and 5, and Supplementary Fig. 6, the [483]SST-GFS/[754]D-E/[798]V-S/[800]A-T/[803]I-R NBCe1 construct was converted from an electrogenic Na$^+$-coupled symporter to an electroneutral anion exchanger with Cl$^-$-driven base flux approximating AE1. These findings demonstrate that a relatively small combination of residues within the structurally flexible NBCe1 ion coordination region contains the essential structural information to encode both the ion transport specificity and the specific mode of transport, i.e., symport vs. exchange function.

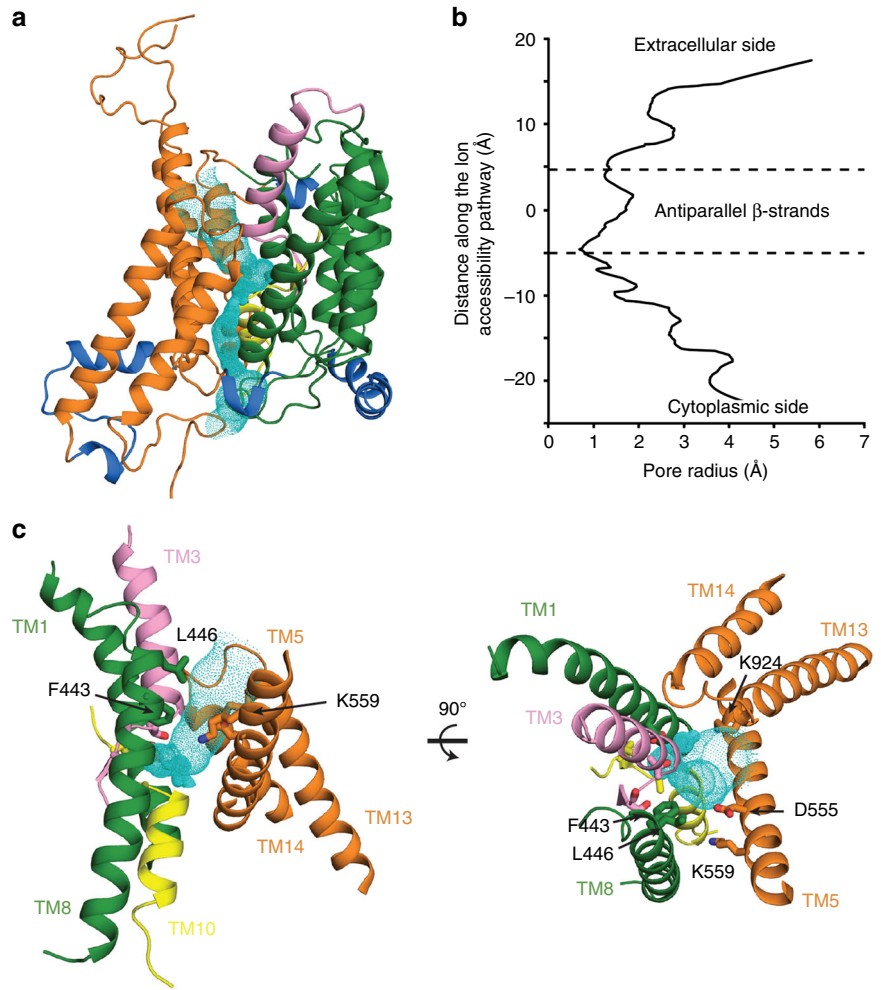

**Fig. 2** Ion accessibility pathway of NBCe1. The ion accessibility pathway (**a**, cyan mesh) and the pore radius value along the pathway (**b**). **c** Selected residues (Phe443, Leu446, Asp555, and Lys559) forming the pathway whose substitution significantly alters NBCe1 function[25–28]. Lys559 is known to bind to DIDS[27], which inhibits NBCe1 function likely via blocking the ion accessibility pathway and/or the conformational changes during the transport cycle

## Discussion

Human NBCe1 is the first eukaryotic Na$^+$-coupled transporter whose near-atomic structure has been solved by cryoEM. The transporter is a member of the anion exchanger family belonging to the APC superfamily, the second largest secondary carrier superfamily[5]. Members of the APC superfamily have structures with internal pseudosymmetry of either 5 + 5 TM or 7 + 7 TM inverted repeats[5]. The 7 + 7 TM inverted repeat fold and domain organization of NBCe1 resemble the structure of AE1[22] and the plant boron transporter Bor1[20] belonging to the SLC4 transporter subfamily, and several unrelated transporters in the nucleobase-ascorbate transporter (NAT) and sulfate permease (SulP) sub-families[31] including the bacterial UraA uracil-H$^+$ symporter[18,32], the bacterial SLC26Dg H$^+$-coupled fumarate symporter SLC26Dg[19], and the fungal UapA purine-H$^+$ symporter[20]. NBCe1 shares several global structural features with these transporters including a gate domain that is smaller than the core domain, two highly tilted antiparallel TM helices (TMs 5 and 12) in the gate domain, and residues in the core domain mediating substrate/ion specificity[33].

Our structure-based mutagenesis of a relatively small combination of residues within the structurally flexible NBCe1 ion coordination region near TM3, TM8, and TM10 helices and the antiparallel β-strands preceding TM3 and TM10, showed that NBCe1 could be converted from an electrogenic Na$^+$-coupled symporter to an electroneutral anion exchanger. To our knowledge, this represents the first example of such a transport mode conversion and indicates that the NBCe1 ion coordination region determines not only its ion transport specificity but also encodes information for its mode of transport. Previous examples of transport mode modulation in general have involved activation of channel-like activity in transporters that function as exchangers. Specifically, a double mutation of residues flanking the central Cl$^-$ in the *E. coli* ClC 2Cl$^-$/H$^+$ exchanger converts the transporter into a channel likely by stabilizing conformations where inner and outer gates are open[34]. The Na$^+$-K$^+$-ATPase (the ATP-dependent Na$^+$/K$^+$-exchanger) is converted into a cation channel when bound to palytoxin likely by uncoupling the inner and outer gates[35]. Finally, naturally occurring mutations in AE1 block its anion exchange activity and uncover a monovalent cation conductance[30,36].

Our cryoEM data demonstrated that NBCe1 is a homodimer confirming previous biochemical, fluorescence energy transfer resonance (FRET), and spatial intensity distribution analysis (SpIDA) data[15,37,38]. The transporters sharing the NBCe1 fold are either dimers[20–22] or in the case of UraA both monomeric and dimeric forms depending on the crystallization conditions utilized[18,32]. Furthermore, SLC26Dg is a monomer in the presence of maltoside detergents, but dimers were also detected in

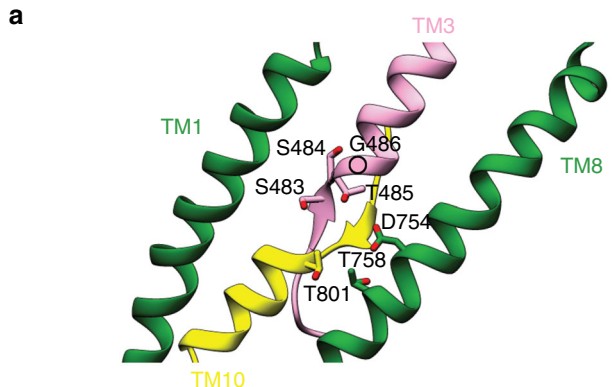

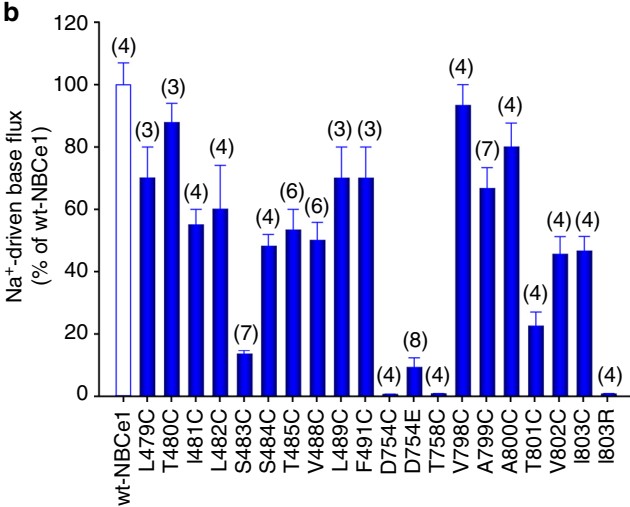

**Fig. 3** NBCe1 ion coordination site. **a** The ion coordination site modeled from the cryoEM densities. Residues important for ion coordination are shown. The circle represents the position of Gly486 in the NBCe1 model. **b** Na$^+$-driven base transport functional mutational analysis of the amino-acid residues located in the ion coordination region. The mean ± s.e.m. is depicted. The number of experiments is shown in parentheses. The residues in this region that are important for ion coordination shown in **a** and those not important for ion coordination (e.g., Leu479, Thr480, Leu489, Phe491, Val798 etc.) are shown for comparative purposes

proteoliposomes[19]. An advantage of dimerization may be that the interaction of two gate domains confers a greater stability in the plane of membrane in the transport cycle.

Unlike channels that have a relatively rigid single conformation, both symporters and exchangers share global structural transitions, e.g., rocker-switch, rocking bundle, elevator mechanisms or a combination of these mechanisms that allow substrates to alternatively have access to their coordination sites from each side of the membrane[33]. For example, the bacterial Na$^+$-coupled leucine transporter (LeuT) has a rocking bundle transport mechanism[39]. LeuT has a 5+5 TM inverted repeat fold and two kinked helices per monomer that are critical for substrate coordination and conformational changes. Transporters with similar LeuT folds including the Na$^+$-coupled glucose symporter vSGLT[40], arginine/agmatine antiporter AdiC[41,42] and the Na$^+$-coupled betaine-glycine symporter BetP[43] have also been predicted to undergo a rocking bundle transport mechanism. Previous studies have suggested that AE1, Bor1, and UapA utilize an elevator transport mechanism[20–22] exemplified by GlpT$_{Ph}$[44]. However, a recent comparison of UraA in the occluded and inward-open states, and the inward-open UapA structure predicted that members of the NAT family undergo a combination

of both elevator and rocking bundle transitions[32]. These findings suggest that determination of additional conformational states of NBCe1 will be required to define its exact transport mechanism.

Our study represents a technical advancement in the cryoEM field given that the structure of the TM region of NBCe1 was solved in the absence of a structured cytoplasmic domain outside the TM region and without the utilization of phase plates[45]. Previous membrane protein structures had structured soluble domains[46–50] or complexed with another soluble protein[45,51–53] to help drive the orientation determination in single-particle cryoEM reconstruction. Importantly, the successful demonstration of the atomic modeling of NBCe1 by analysis of only its TM region opens the prospect for obtaining high-resolution structures of numerous smaller membrane proteins. In addition, the ability of cryoEM to sort particles from a single heterogeneous dataset[54,55] will be advantageous to potentially resolve multiple conformational cryoEM structures of NBCe1.

## Methods

**Expression and purification of NBCe1**. Human N-terminally Strep(II)-tagged wild-type (wt) NBCe1-A in the pTT vector (Addgene) was transfected into HEK-293 cell (ATCC) monolayers using the calcium phosphate method. The cells were grown in Dulbecco's Modified Eagle's medium (Thermo Fisher Scientific) with 5% fetal bovine serum (Thermo Fisher Scientific) on 10-cm plates for ~24 h. The transfected cells from ~200 plates were pelleted by centrifugation at 2000 × $g$ for 10 min. The cell pellets were solubilized with 1% Triton X-100 in buffer A (50 mM Tris-HCl, pH 7.5, 500 mM NaCl) supplemented with complete protease inhibitor cocktail (Thermo Scientific) for 30 min. Detergent insoluble material was removed by centrifugation (20,000 × $g$ × 30 min) and the supernatant was loaded onto the 5-ml StrepTrap HP column (GE Healthcare). The column was washed with buffer A containing 0.01% n-dodecyl α-D-maltopyranoside (DDM, Anatrace), and bound protein was eluted with the same buffer containing 0.03% DDM supplemented with 2.5 mM D-desthiobiotin.

For preparation of samples for cryoEM, NBCe1-A was mixed with amphipol PMAL-C8 (Anatrace) at 1:3 (w/w) dilution with gentle agitation overnight at 4 °C. Detergent was removed with Bio-Beads SM-2 (Bio-Rad) incubated with samples for 1 h at 4 °C, and the beads were subsequently removed by centrifugation (2000 × $g$ × 5 min). Amphipol containing protein was further purified on a Superose 6 column (GE Healthcare) in 20 mM Tris-HCl, pH 7.5, 150 mM NaCl. The peak corresponding to dimeric NBCe1-A was collected for cryoEM analysis.

**Electron microscopy sample preparation and imaging**. For electron microscopy of negatively stained protein, 2 μl of NBCe1-A sample (~0.1 mg ml$^{−1}$) was applied to a glow-discharged EM grid covered with a thin layer of carbon film. After 10 s incubation, the grid was stained with 0.8% uranyl formate. EM micrographs were recorded on a TIETZ F415MP 16-megapixel CCD camera at 50,000 × nominal magnification in an FEI Tecnai F20 electron microscope operated at 200 kV. Micrographs were saved by 2 × binning to yield a calibrated pixel size of 4.41 Å.

For cryoEM, 3 μl of NBCe1-A (~0.4 mg ml$^{−1}$) was applied to a glow-discharged Quantifoil 300-mesh R1.2/1.3 grid. The grid was blotted with filter paper to remove excess sample and flash-frozen in liquid ethane with FEI Vitrobot Mark IV. The frozen-hydrated grids were loaded into an FEI Titan Krios electron microscope operated at 300 kV for automated image acquisition with Leginon[56]. Micrographs (dose-fractionated movies) were acquired with a Gatan K2 Summit direct electron detection camera operated in the super-resolution mode (7676 × 7420 pixels) at a calibrated magnification of 36,764 × and defocus values ranging from −1.4 to −3.2 μm. A GIF Quantum LS Imaging Filter (Gatan) was installed between the electron microscope and the K2 camera with the energy filter (slit) set to 20 eV. The dose rate on the camera was set to ~8 e$^−$ pixel$^{−1}$ s$^{−1}$ and the total exposure time was 12 s fractionated into 48 frames of images with 0.25 s exposure time for each frame. A total of 3378 micrographs were collected.

**Image processing**. The frame images of each micrograph were aligned and averaged for correction of beam-induced drift using MotionCor2[17]. The local motion within a micrograph was corrected using 5×5 patches. Two average images, with and without dose-weighting, from all except the first frame were generated with 2×binning (final pixel size of 1.36 Å on the sample level) for further data processing. A total number of 3136 good micrographs were picked for image processing by visual inspection of the average images and power spectra after the drift correction.

The defocus values of the micrographs were measured on the dose-unweighted average images by CTFFIND4[57]. The dose-weighted average images were used for particle picking and subsequent image processing. A total of 1,729,419 particles were automatically picked using Gautomatch [K. Zhang, MRC LMB (www.mrc-lmb.cam.ac.uk/kzhang/)] and boxed out in 192 × 192 pixels.

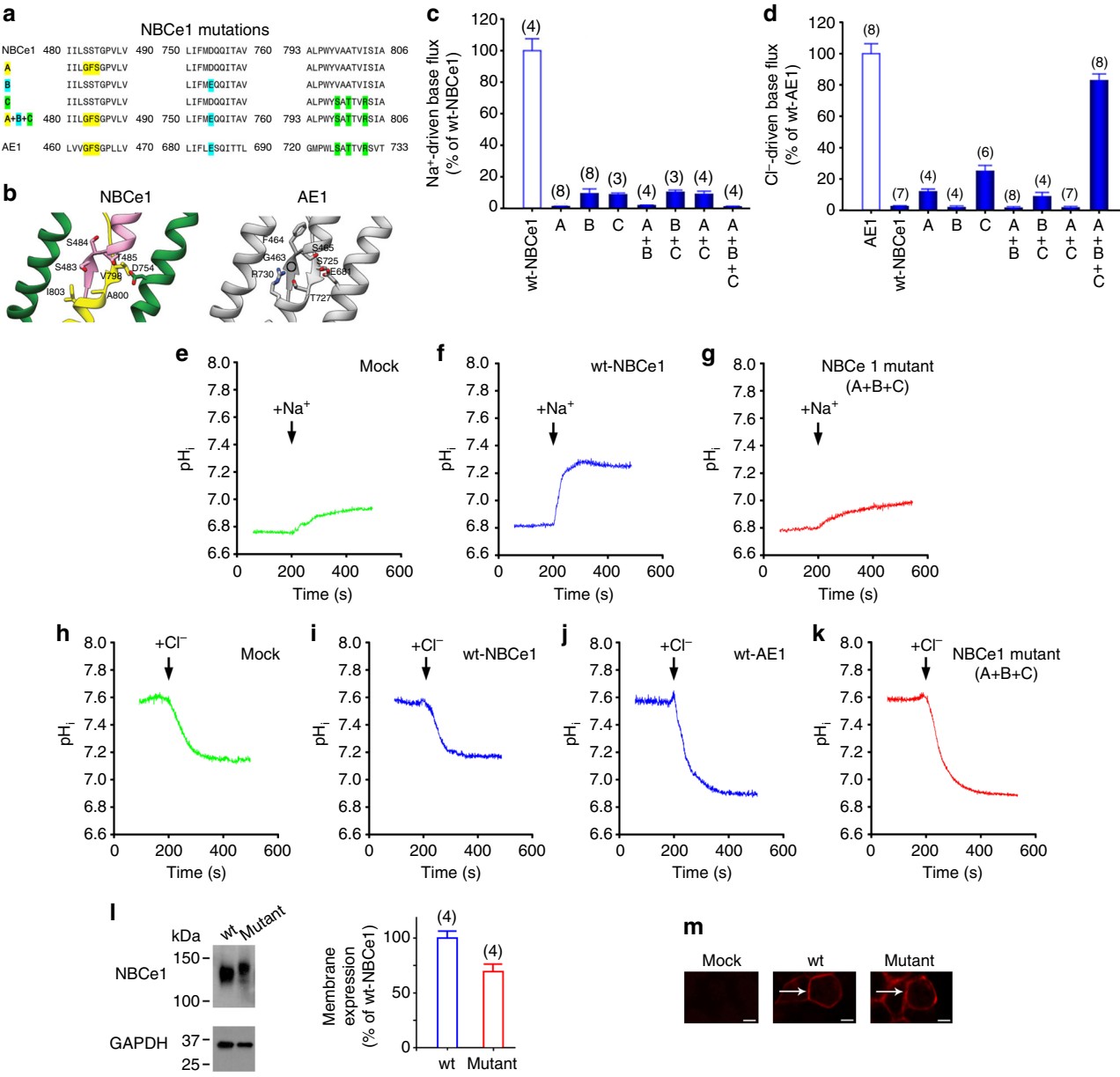

**Fig. 4** Conversion of NBCe1 into an anion exchanger. **a** NBCe1 and AE1 amino-acid alignment showing location of the NBCe1 residues replaced with the corresponding AE1 residues. **b** Location of the NBCe1 residues replaced with the corresponding AE1 residues located in ion coordination site regions of NBCe1 and AE1[22] (PDB: 4YZF) formed by antiparallel β-strands preceding TMs 3 and 10, and in TM 8 and 10 are shown. **c, e–g** Loss of Na$^+$-driven base transport and **d, h–k** gain of Cl$^-$-driven base transport in the mutant constructs. **l, m** Cell surface expression. Immunoblot analysis of sulfo-NHS-SS-biotin labeled plasma membrane proteins in HEK-293 cells-expressing wt-NBCe1 and NBCe1 mutant (A+B+C). **l** Cell surface expression was normalized to GAPDH (mutant vs. wt, $p < 0.05$; one-way ANOVA followed by Tukey's test). **m** Immunocytochemistry of HEK-293 cells showing plasma membrane localization of the wt and NBCe1 mutant (A+B+C) transporters, scale bars 5 μm. **c, d, l** The mean ± s.e.m. is depicted. The number of experiments is shown in parentheses

The particles were first subjected to 3D classifications by GPU-accelerated RELION-2[58,59] using an oval-shaped disk low-pass filtered to 60 Å as the initial model. The particles were separated into six classes for 69 iterations and the best class contained 687,322 particles, which were then sent to a second run of 3D classification to be sorted into five classes. A number of 214,196 particles were found in the best class after 60 iterations of the second run of 3D classification. A twofold symmetry was enforced in the above 3D classifications. 3D classifications without symmetry (i.e., C1) were also tested but no further improvement on the separation of heterogeneity was found. The 214,196 particles were then sent to 2D classification using RELION-2 and sorted into 200 classes for 72 iterations.

A total number of 184,561 particles in 134 good classes that show discernible high-resolution features such as α-helices were combined for 3D auto-refinement. The 3D auto-refinement was performed using a spherical mask (180 Å in diameter) by RELION-2. The resolution was estimated to be 3.9 Å by relion_postprocess in RELION-2 with a soft auto-mask using the "gold-standard" FSC at 0.143 criterion.

The final cryoEM map was sharpened with B-factor and low-pass filtered to the stated resolution using RELION-2. The local resolution was calculated by ResMap[60] using two cryoEM maps to independently refine from halves of the data.

**Model building and refinement and structure visualization**. The overall 3.9 Å cryoEM structure was at a sufficient resolution to build a de novo atomic model for TMs 1–3, TMs 5–14, amphipathic helices H1–4 and short (single turn) cytoplasmic helix H5 in Coot[61].

Aromatic residues in many of these TMs were clearly visible in our cryoEM structure and were used as landmarks for our model building, especially in areas where multiple aromatic residues are in close proximity. TMs 1–3 along with the loop between each helix were built de novo starting at residues Phe431-Tyr433 and continued with side chain placements into the protrusions visible in the helical densities. Aromatic residues including Phe443, Phe461, Phe471, Phe474, Phe791,

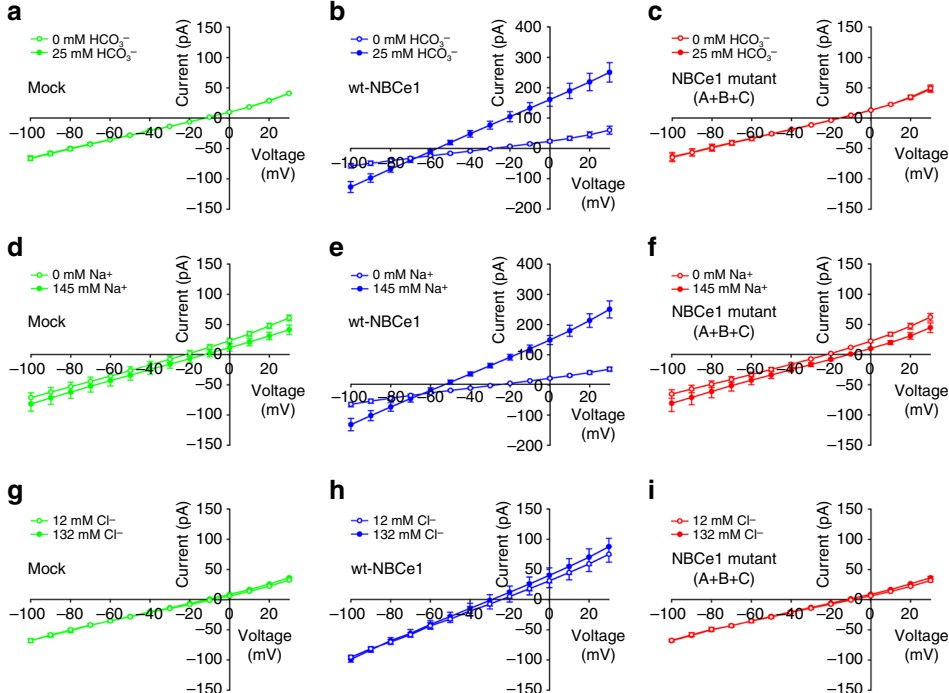

**Fig. 5** Electrophysiological characterization of the NBCe1 $^{483}$SST-GFS/$^{754}$D-E/$^{798}$V-S/$^{800}$A-T/$^{803}$I-R mutant (A+B+C). The mutant $^{483}$SST-GFS/$^{754}$D-E/$^{798}$V-S/$^{800}$A-T/$^{803}$I-R construct (NBCe1 Mutant A+B+C), which lost Na$^+$-driven base transport and whose magnitude of Cl$^-$-driven base transport approximated AE1, was studied electrophysiologically using whole-cell patch clamping. Unlike electrogenic wt-NBCe1 (**b**, **e**, **h**), the measured I–V curves of the mutant construct (**c**, **f**, **i**) under the various study conditions did not differ from those obtained in mock-transfected cells (**a**, **d**, **g**) demonstrating that the mode of transport of the mutant construct was electroneutral

Phe496, and Phe498 provided checkpoints for building TMs 1–3. The side chain positions were temporarily refined to the cryoEM helical densities using the real-space refine zone feature in Coot with planar peptide, Ramachandran and α-helix restraints. The loops between the TMs were similarly refined in Coot without the α-helix restraints. This procedure was repeated for helices TMs 5–14. TM5 was built in reverse from C- to N-terminus starting at aromatic residues Tyr666–Tyr667 and guided by residues Phe539, Phe544, Phe550, Phe552, Tyr554, and Phe557. TMs 6 and 7 were built starting with residues Tyr674-Phe675, located at the small loop region between the two helices. CryoEM densities for Phe653, Phe656, Tyr660, Phe669, Phe686, and Phe695 supported our model building for TMs 6 and 7. Model building for TMs 8–10 and the loops originated from residues Tyr775 and His776 was done with guidance from residues Trp781, Trp796, Tyr797, and His807. TMs 11–14 were built starting with residues Tyr862, Phe865, and Tyr867 from TM12. Aromatic residues located in both the helices and loop regions between TMs 11–14 aided the model building of these regions. TM4 was slightly challenging to model as some of the aromatic residue densities were missing. Using Phyre2[62] to predict the secondary structure based on the sequence of TM4, we were able to model this helix with guidance from residues Trp512, Trp516, and Phe519. Helix H1 contains two sites of consecutive aromatic residues (Phe112-Phe113 and Phe117-Tyr118) that were used as starting points for de novo model building. Helices H2–H4 and cytoplasmic helix H5 were simultaneously built during the model building of the other TMs.

In addition to the TMs, we were able to build a model for a portion of EL3 located between TMs 5 and 6. A polyalanine chain was built between residues Tyr566-Ser582 and Val638-Asp647 of EL3 since we were able to accurately trace the backbone for this dynamic loop.

The monomeric model of NBCe1-A was duplicated across the twofold symmetry in Chimera[63] and saved as one dimeric model. The dimeric model of NBCe1-A was subjected to further global refinement with simulated annealing using the real-space refinement (RSR) feature in the PHENIX software package[64]. After five iterations in RSR with simulated annealing, the model was further analyzed in MolProbity[65]. Residues with poor rotamer or considered a Ramachandran outlier were fixed in Coot. Another round of RSR and MolProbity analysis were performed. The Molprobity score of the final atomic model was 2.11 with a 9.23 clashscore, and without Ramachandran outliers, Cβ deviations, bad bonds, poor rotamers, and cis-peptides. The cross-correlation coefficient between the final atomic model and the cryoEM density calculated from RSR is 0.801. The structure is visualized in UCSF Chimera[63]. Protein sequences were aligned using Clustal Omega[66].

The final NBCe1-A atomic model was superposed onto AE1[22] (PDB: 4YZF) model using the superpose command from the CCP4 package[67]. The superimposed figures between NBCe1-A and AE1 were displayed in UCSF Chimera. Permeation pathway analysis was calculated using the HOLE2 software[23] and displayed in Pymol[68]. The electrostatic analysis was calculated using APBS[69] once the model was modified in PDB2PQR[70]. The electrostatic map was visualized in UCSF Chimera.

**Structure-guided mutagenesis.** Mutations were introduced in wt-NBCe1-A using the QuikChange Lighting Site-Directed Mutagenesis Kit (Agilent Technologies; Supplementary Table 1). Twenty-seven individual constructs were studied. All sequences were verified using the University of California Los Angeles Genotyping and Sequencing Core using the BigDye terminator kit version 3.1 (Invitrogen Life Technologies) and resolved with a 3730 XL ABI sequencer (Applied Biosystems, Life Technologies).

**Cell surface biotinylation and immunocytochemistry.** The constructs were initially assessed for plasma membrane expression using sulfo-NHS-SS-biotin plasma membrane labeling and immunocytochemistry using a previously characterized rabbit polyclonal NBCe1-A antibody[25]. Twenty-four hours following transfection of HEK-293 cells with wt-NBCe1-A and various NBCe1-A mutants using Lipofectamine 2000 (Thermo Fisher Scientific), plasma membrane proteins were labeled using sulfo-NHS-SS-biotin (Pierce) and pulled down using streptavidin-agarose resin according to the manufacturer's protocol. Immunoblots of plasma membrane pulled down proteins were probed with the NBCe1-A antibody (1:10,000 dilution) and cell lysates were probed with a GAPDH (A-3) antibody (sc-137179, Santa Cruz Biotechnology; 1:5000 dilution). Cell surface expression was normalized to GAPDH. The experiments were done in triplicate. All constructs except one ($^{483}$SST-GFS/$^{754}$D-E/$^{798}$V-S/$^{800}$A-T/$^{803}$I-R) had normal membrane expression (Fig. 4l) For immunocytochemistry (Fig. 4m), 24 h post-transfection cells grown on coated coverslips were permeabilized with methanol (room temperature), washed with phosphate-buffered saline (PBS) and incubated with the NBCe1-A antibody (1:100 dilution in PBS; room temperature for 30 min). Following another PBS wash, the coverslips were incubated with Alexa Fluor 594 goat anti-rabbit secondary antibody (30 min at room temperature; 1:500 dilution in PBS; Thermo Fisher Scientific). The antibody was washed off with PBS and fluorescence images of the cells were captured with a PXL charge-coupled device camera (model CH1; Photometrics) coupled to a Nikon Microphot-FXA epi-fluorescence microscope (Melville).

**Base transport activity assays**. HEK-293 cells grown on coated coverslips were transfected with wt-NBCe1-A, various NBCe1-A mutants, and wt-AE1 and studied 24 h later. Intracellular pH ($pH_i$) was monitored with a microscope-fluorometer in cells loaded at room temperature for ~20 min with the fluorescent intracellular pH ($pH_i$) probe BCECF (using esterified BCECF-AM; Life Technologies). The intracellular fluorescence (excitation ratio 500 nm/440 nm; emission 530 nm) was acquired every 0.5 s and calibrated at the end of each experiment with valinomycin (Sigma-Aldrich) and nigericin (Sigma-Aldrich). To determine the $Na^+$-driven base flux, the initial rate of change of $[H_{in}^+]$ ($d[H_{in}^+]/dt$) was measured using a linear curve fit in the first 5–10 s following the increase in bath $Na^+$ from zero to 140 mM. A similar analysis was done in the $Cl^-$-driven base flux protocol following the increase in bath $Cl^-$ from zero to 119 mM. Base flux was calculated as $d[H_{in}^+]/dt$ x $\beta_T$, where the total cell buffer capacity ($\beta_T$) is equal to the intrinsic buffer capacity ($\beta_i$) plus the $HCO_3^-$-buffer capacity. The data were background (mock-transfected, i.e., empty plasmid) subtracted and depicted as percentage of the wild-type flux.

$Na^+$-driven base flux: The cells were initially bathed in $Na^+$-free buffer: 115 mM TMACl, 2.5 mM $K_2HPO_4$, 1 mM $CaCl_2$, 1 mM $MgCl_2$, 5 mM glucose, 25 mM TMA-$HCO_3$, pH 7.4, and 30 μM EIPA (to block endogenous $Na^+/H^+$ exchange) until a steady-state was achieved. Following the addition of $Na^+$ (115 mM NaCl, 2.5 mM $K_2HPO_4$, 1 mM $CaCl_2$, 1 mM $MgCl_2$, 5 mM glucose, 25 mM $NaHCO_3$, pH 7.4, with 30 μM EIPA), the $Na^+$-driven base flux was measured. The data are the mean of 3−8 experiments.

$Cl^-$-driven base flux: The cells were initially bathed in a $Cl^-$-free buffer containing 115 mM Na gluconate, 2.5 mM $K_2HPO_4$, 7 mM Ca gluconate, 1 mM Mg gluconate, 5 mM glucose, 25 mM $NaHCO_3$, pH 7.4, and 30 μM EIPA. After a steady-state was achieved, a $Cl^-$-containing buffer (115 mM NaCl, 2.5 mM $K_2HPO_4$, 1 mM $CaCl_2$, 1 mM $MgCl_2$, 5 mM glucose, 25 mM $NaHCO_3$, pH 7.4, and 30 μM EIPA) was added and the $Cl^-$-driven base flux was measured. The data are the mean of 4−8 experiments.

**Electrophysiology assays**. Whole-cell patch clamping was used to compare the electrogenicity of wt-NBCe1-A and an $^{483}$SST-GFS/$^{754}$D-E/$^{798}$V-S/$^{800}$A-T/$^{803}$I-R mutant expressed in HEK-293 cells as described[28]. Steady-state currents were measured using a holding potential of −55 mV, with a series of 400-ms voltage pulses (increment of 10 mV from −100 to + 30 mV). The data are the mean of 3–10 experiments.

$HCO_3^-$-dependent current: The patch solution contained 125 mM CsOH, 10 mM NaOH, 100 mM gluconic acid, 1 mM $CaCl_2$, 10 mM TEACl, 10 mM EGTA, 10 mM HEPES, pH 7.4, and the bath solutions contained 120 mM NaCl, 1.5 mM $CaCl_2$, 10 mM CsCl, 10 mM HEPES, pH 7.4, with 25 mM Na gluconate or 25 mM $NaHCO_3$ (replacing 25 mM Na gluconate).

$Na^+$-driven current: The patch solution contained 125 mM CsOH, 10 mM NaOH, 100 mM gluconic acid, 1 mM $CaCl_2$, 10 mM TEACl, 10 mM EGTA, 10 mM HEPES, pH 7.4, and the bath solutions contained 1.5 mM $CaCl_2$, 10 mM CsCl, 10 mM HEPES, pH 7.4, with 120 mM TMACl and 25 mM choline $HCO_3$, or 120 mM NaCl (replacing 120 mM TMACl) and 25 mM $NaHCO_3$ (replacing 25 mM choline $HCO_3$).

$Cl^-$-driven current: The patch solution contained 105 mM CsOH, 70 mM gluconic acid, 1 mM $CaCl_2$, 10 mM TEACl, 10 mM EGTA, 10 mM HEPES, 10 mM $NaHCO_3$, 15 mM choline $HCO_3$, pH 7.4, and the bath solutions contained 1.5 mM $CaCl_2$, 9 mM CsCl, 10 mM HEPES, 10 mM $NaHCO_3$, 15 mM choline $HCO_3$, pH 7.4, with 120 mM Na gluconate or 120 mM NaCl (replacing 120 mM Na gluconate).

**Data availability**. The final cryoEM density map of human NBCe1-A has been deposited to the Electron Microscopy DataBank (EMDB) under the accession code EMD-7441. The final atomic model was deposited into the Protein Data Bank (PDB) under the accession code 6CAA. All other relevant data are available in this article and its Supplementary Information files, or from the corresponding authors upon reasonable request.

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

## Acknowledgements

This work was supported by funds from the NIH (DK077162, DK007789, GM071940, DE025567, and AI094386), the Allan Smidt Charitable Fund, the Factor Family Foundation and the Ralph Block Family Foundation. We acknowledge the Electron Imaging Center for NanoMachines for the use of their instruments supported by the University of California at Los Angeles and instrumentation grants from both the NIH (1S10RR23057 and 1U24GM116792) and the NSF (DBI-1338135).

## Author contributions

I.K., A.P. and Z.H.Z. were responsible for the initiation and the overall management of the project. N.A., D.N., and L.K. performed cloning, mutagenesis, expression in HEK-293 cells and biochemical characterization of wt-NBCe1 and mutants. K.T. purified wt-NBCe1 protein from transfected HEK-293 cells for cryoEM. I.K. designed the functional studies that were performed by R.A., L.K., and X.S. J.J. and Z.H.Z. designed the cryoEM strategies. J.J. obtained the cryoEM data and performed image analyses. K.W.H., J.J., and Z.H.Z. performed model building and structural interpretation. I.K., Z.H.Z., A.P., and K. W.H. wrote the manuscript.

## Additional information

**Competing interests:** The authors declare no competing interests.

