## [Peer Review File · Nature Communications]

Reviewers' Comments:

Reviewer #1:

Remarks to the Author:

In this work, the authors determined the structure of human sodium-coupled acid-base transporter NBCe1 by cryo electron microscopy at 3.9 angstroms resolution. The relatively high-resolution structure of NBCe1 provides the framework for sodium-coupled acid-base and sodium-coupled SLC4 transporters for functional study. Intriguingly, NBCe1 exhibits similar structural fold with several reported transporters such as E. coli uracil:proton symporter UraA and anion exchanger AE1.

The authors identified the key residues in the ion accessibility pathway and the coordination site. Mutagenesis of these key residues hampers the substrate transport. Two disease-related mutants are also found to be located at the ion coordination site. Thus, this structure gives a potential structural explanation on how these disease-related mutants affect the normal function.

Through structural alignment of NBCe1 and AE1, the author mutated specific residues in NBCe1 to the corresponding residues in AE1. These mutations alter NBCe1 to mimic the function of an anion exchanger. Such functional conversion indicates that some symporter and exchanger may share comparable transport machinery with subtle differences at the substrate binding sites.

Overall this is a high-quality structure of a transporter with physiological and pathological significance. In particular, it is the FIRST cryo-EM structure for any SLC members with resolution above 4 Å, representing a milestone in the structural biology of SLC proteins. No doubt, this work is qualified for publication with Nature Communication before some minor points are addressed.

1. Line 20, please add SLC23 to "SLC4, SLC9 21 and SLC26".
2. The author mentioned that NBCe1 resembles some similar structure features to unrelated structures. It would be better to discuss the differences between NBCe1 and the reported structures, including the recently published dimeric structure of E. coli uracil:proton symporter (Yu et al, 2017, Cell Research, 27:1020).
3. In the Supplementary Fig. 4b, the authors showed the model of NBCe1 with DIDS. However, one cannot tell from the manuscript whether this is a determined structure or a computational model. It should be clarified.
4. In Fig. 3b, the biochemical data of some residues (e.g. L497, T480.....) have been shown. However, these mutations are not mentioned in the main-text or the figure legend. In addition, biochemical characterization of the disease related mutations may be discussed in more details, citing the previous studies analyzing these mutations.

Reviewer #2:

Remarks to the Author:

Kurtz NBCe1 Structure

SLC transporters are among the most poorly studied of all human gene products. This paper provides the first (cryoEM) structure of a human sodium-dependent transporter. Cryo-EM to 3.9 Å for a 130 kDa membrane protein is certainly at the technical forefront and should be highlighted. The 7+7 TM inverted repeat structure places NBCe1 in a special class of 14 TM fold membrane proteins, that include other members of the SLC4 bicarbonate transporter family like SLC4A1, the human anion

exchanger (Band 3), and the members of the SLC26 anion transport family. The structure allowed the discovery and positioning of key residues in the substrate binding site and the localization of disease-causing mutations. The dimer state of the protein is confirmed by the structure. What is quite exciting is the ability to convert NBCe1 from a co-transporter into an exchanger by mutating a few residues at the substrate-binding site into those found in SLC4A1. The paper is clearly written and easy to read, and is a significant advance to the structural biology of SLC membrane transport proteins.

Minor suggestions:

1. I would include ...Human SLC4A4 Sodium-coupled... in the title.
2. SLC4 and SLC26 belong to a novel class of 14TM segment inverted 7+7 fold of membrane proteins highlighted in a 2017 review by Chang and Geertsma, which could be cited. Similarly, a definitive 2016 review of human SLC4A1 (Band 3) by Reithmeier et al., which includes a comparison with SLC4A4 could be cited.
3. DIDS may block the ion accessibility site but may also block the conformational change associated with transport as this inhibitor is located at the interface between the gate and core domains.
4. There is lots of evidence for a dimeric structure in human SLC4A1.
5. The density (Fig S2) looks sufficiently detailed to identify bulky side chains and thus put the sequence into register in the map. However, most of the side chains do not appear to be resolved. Thus, the model can identify other side chains but perhaps shouldn't show their orientations as that information is not experimentally determined (i.e. side chains should be labelled with their identity but modelled as polyalanine in the PDB file). At a minimum, this limitation should be clearly stated.
6. One aspect of the methodology is worrying: why was the FSC estimated with a soft spherical mask and not the standard close-fitting mask default of Relion with correction of the FSC for the mask. It is essential to show that the mask (even though described as loose fitting and soft edged) does not cut into the protein density and thereby inflate the resolution estimate by FSC.
7. It would be useful to relate the effect of the individual cysteine mutations on transport to the conversion to an anion exchanger with the triple mutant. For example, some single mutations like D754E inhibit co-transport but do not yet demonstrate anion exchange.
8. It might also be useful to briefly contrast the NBCe1 structure with SLC5 sodium-dependent sugar transporters like vSGLT.
9. The section at the end of the Discussion on the advances in single particle cryo EM of membrane proteins could be abbreviated or eliminated all-together. The use of amphipol PMAL-C8 is however, novel and interesting (closest previously has been amphipol A8-35). This may be a better place to more fully discuss the nature of the sodium and bicarbonate (anion) binding sites and the effect of disease-causing mutations on protein structure.

Reviewer #3:

Remarks to the Author:

NCOMMS-17-20257

CryoEM Structure of the Human Sodium-Coupled Acid-Base Transporter NBCe1

Kevin W. Huynh^{1,2*}, Jiansen Jiang^{2,3*}, Natalia Abuladze^{1*}, Kirill Tsurulnikov^{1*}, Liyo Kao¹, Xuesi Shao⁴, Debra Newman¹, Rustam Azimov¹, Alexander Pushkin¹, Z. Hong Zhou^{2,3}, and Ira Kurtz^{1,5}

Overall

This is a really wonderful structural biology study elucidating the membrane structure of human NBCe1 at 3.9Å by cryoEM. The authors have also made some very intriguing observations concerning a Na⁺ binding site and the anion (Cl⁻) exchange site for other SLC4 proteins (e.g., AE1). The interdigitated TM-segments while expected from the AE1 crystal structure seem better resolved by this solution rather than crystallization method. The weakest part of this entire manuscript is currently the ion transport data. This does not appear to be due to inappropriateness of the experiments but rather

in the authors choices of what data to include as part of the manuscript (see detailed comments). Overall this is a very complete study, but the authors would be well served by highlighting specific experiments.

Major Points

1) Abstract / Introduction

a) Slc4 family is typically referred to as the "HCO₃ transport" family. Why do the authors refer to this NBCe1 protein as a Na⁺ coupled CO₃²⁻ symporter? This does not even seem to match the acronym.

b) Including Slc9 as a Na⁺ coupled acid-base transport seems odd as other Na⁺ coupled-base transport systems are not included, e.g., Slc5 monocarboxylate transporters, Slc10 Na⁺ bile acid transporters, Slc13 Na⁺ coupled sulfate / carboxylate transporters, Slc23 Na⁺ couple ascorbic acid transporters, Slc34 Na⁺ coupled phosphate transporters, etc

2) Na⁺ coupling

a) The authors have an attractive speculation for the molecular determinants of Na⁺ coupled vs. H⁺ residues (li 115-117).

b) The evolutionary reference seems misplaced as the authors are presuming that the Slc4 anion exchangers are the most evolutionarily distant and the Na⁺ coupling of some Slc4-transporters is more recent. To my knowledge there is no evidence for assigning what the Slc4 proto-transporter may have transported. So it could be equally likely that the Na⁺ coupling "degenerated" or changed to protonation.

3) Figure 4 – transport data

a) Base-flux. Representative traces for 4b & 4c should be shown in Supplemental data so that readers see the primary data and know how the authors analyze the BCECF data. Please also indicate how the base-flux data (as well as Na⁺ and Cl⁻ dependent components) are quantified (e.g., linear or non-linear curve fit)

b) Na⁺ dependent currents (4h,i, j) are very small and have different reversal potentials (compare 4i to 4f). Why do 4f and 4i (solid symbols) differ by 4-fold. This should be the same experimental condition, i.e., full Na⁺ with 25 mM HCO₃⁻. Perhaps since there seems to be wide variation between preparations, the authors should show individual experiments (in the presence and absence of HCO₃⁻) ± Na⁺ and ± Cl⁻. The authors point that NBCe1 is a Na⁺ coupled HCO₃⁻ (or CO₃²⁻) symporter is difficult to understand without clearer experimental records.

c) Cl⁻ dependent currents (4k, l, m). presumably as in "Na⁺ dependence," these are experiments in the presence of 25 mM HCO₃⁻. This is not specified. If k,l,m are not HCO₃⁻ solutions, we do not learn anything from the experiments.

d) Surface protein expression. Figure 4 mutational analysis relies on mutants A, B, C and A+B+C all trafficking to the plasma membrane to the same extent as wt-NBCe1. Here there is not data to indicate that this is true. As the authors have changed the NBCe1 amino acids to AE1 amino acids, it is implied that there is no difference in cell surface expression. Without knowing this plasma membrane expression and the comparative amount of expression, one cannot interpret most of the data plotted in figures 4c-m. An additional supplement showing amount of surface expression of A+B+C v wt-NBCe1 is minimally needed.

4) Discussion

a) (li 145-147) While the investigators have a unique structure and a novel method applied to SLC4 proteins, claims that this is the "first" report of a SLC4 family member as a dimer are not true. The early crystal structure of AE1 [Wang DN, Kuhlbrandt W, Sarabia VE, and Reithmeier RA. Two-dimensional structure of the membrane domain of human band 3, the anion transport protein of the erythrocyte membrane. *Embo J* 12: 2233-2239, 1993] showed AE1 as a dimer as well as a dimer of dimers. In 2015, Arakawa and Iwata provided a 3.5 Å structure of AE1 also illustrating a dimer [Arakawa T, Kobayashi-Yurugi T, Alguel Y, Iwanari H, Hatae H, Iwata M, Abe Y, Hino T, Ikeda-Suno C, Kuma H, Kang D, Murata T, Hamakubo T, Cameron AD, Kobayashi T, Hamasaki N, and Iwata S. Crystal structure of the anion exchanger domain of human erythrocyte band 3. *Science* 350: 680-684,

2015].

b) (li 162 - 164) The authors have made an important contribution, but they have overstated their result by indicating they are modeling a 130 kD NBCe1 protein. In previous parts of the manuscript, the authors are careful to indicate that just the membrane and near membrane regions are modeled (li 67-70). This include the cytoplasmic N-terminal Met1-Gly400 (~30-40 kD) and the cytoplasmic C-terminus Asp966-Cys1035 (~8-9kD). In fact, lines 69-70 explicitly states that "...due to the dynamic nature of these regions and were not modeled..." Please state what was actually reported. It seems that ~80 kD of the membrane core of the protein was modeled. There are certainly ways to phrase the comment that this is a big protein without claiming to model the entire protein.

c) Perhaps the authors could better discuss the merits of cryoEM structure(s) of membrane proteins v traditional xray crystallography. To me, the cryoEM structure of a protein in solution is highly preferred to a static crystal.

5) Methods

a) (li 283-284) Please indicate if wt-NBCe1-A mutations were sequence verified.

Minor points

- (li 21) Define "APC protein superfamily"
- Might be worth citing some of the actual NBCe1 literature other than just the authors published works.

Summary of responses and changes in the revised manuscript/figures

We thank the three Reviewers for their precious time and their invaluable critiques, which are
addressed in this revised manuscript.

The Reviewers' comments are in black and our responses are in blue.

Reviewer #1 (Remarks to the Author):

In this work, the authors determined the structure of human sodium-coupled acid-base transporter
NBCe1 by cryo electron microscopy at 3.9 angstroms resolution. The relatively high-resolution
structure of NBCe1 provides the framework for sodium-coupled acid-base and sodium-coupled SLC4
transporters for functional study. Intriguingly, NBCe1 exhibits similar structural fold with several
reported transporters such as *E. coli* uracil:proton symporter UraA and anion exchanger AE1.

The authors identified the key residues in the ion accessibility pathway and the coordination site.
Mutagenesis of these key residues hampers the substrate transport. Two disease-related mutants are
also found to be located at the ion coordination site. Thus, this structure gives a potential structural
explanation on how these disease-related mutants affect the normal function.

Through structural alignment of NBCe1 and AE1, the author mutated specific residues in NBCe1 to
the corresponding residues in AE1. These mutations alter NBCe1 to mimic the function of an anion
exchanger. Such functional conversion indicates that some symporter and exchanger may share
comparable transport machinery with subtle differences at the substrate binding sites.

Overall this is a high-quality structure of a transporter with physiological and pathological significance.
In particular, it is the FIRST cryo-EM structure for any SLC members with resolution above 4 Å,
representing a milestone in the structural biology of SLC proteins. No doubt, this work is qualified for
publication with Nature Communication before some minor points are addressed.

We thank Reviewer 1 for providing us with insightful comments and suggestions. We are pleased that
Reviewer 1 views our manuscript as deserving to be in *Nature Communications*. We have addressed
the Reviewer's comments in our revised manuscript.

1. Line 20, please add SLC23 to "SLC4, SLC9 21 and SLC26".

As suggested, we now reference these and additional SLC transporters in the revised manuscript.

2. The author mentioned that NBCe1 resembles some similar structure features to unrelated
structures. It would be better to discuss the differences between NBCe1 and the reported structures,
including the recently published dimeric structure of *E. coli* uracil:proton symporter (Yu et al, 2017,
Cell Research, 27:1020).

We have expanded our Discussion section as requested to include a comparison between the other
reported structures including the recently published dimeric structure of *E. coli* uracil:proton
symporter, which we now refer to Yu et. al. 2017 (ref. 32).

3. In the Supplementary Fig. 4b, the authors showed the model of NBCe1 with DIDS. However, one
cannot tell from the manuscript whether this is a determined structure or a computational model. It
should be clarified.

Our cryoEM structure was determined in the absence of DIDS. In Supplementary Fig. 4b, we utilized
previous functional inhibition data (ref. 27) and superimposed our atomic model onto the AE1

structure that was resolved in the presence of DIDS (PDB: 4YZF) using the superpose command from
the CCP4 package. We have modified the Supplementary Fig. 4 legend to clarify this point as
requested.

4. In Fig. 3b, the biochemical data of some residues (e.g. L497, T480.....) have been shown.
However, these mutations are not mentioned in the main-text or the figure legend. In addition,
biochemical characterization of the disease related mutations may be discussed in more details, citing
the previous studies analyzing these mutations.

The Fig. 3b legend has been modified as requested. Regarding the patient mutations residing in the
ion coordination site, we prefer at this juncture not to add further details in addition to what was
previously discussed in the Results section.

Reviewer #2 (Remarks to the Author):

Kurtz NBCe1 Structure

SLC transporters are among the most poorly studied of all human gene products. This paper provides
the first (cryoEM) structure of a human sodium-dependent transporter. Cryo-EM to 3.9 Å for a 130
62 kDa membrane protein is certainly at the technical forefront and should be highlighted. The 7+7 TM
inverted repeat structure places NBCe1 in a special class of 14 TM fold membrane proteins, that
include other members of the SLC4 bicarbonate transporter family like SLC4A1, the human anion
exchanger (Band 3), and the members of the SLC26 anion transport family. The structure allowed the
discovery and positioning of key residues in the substrate binding site and the localization of disease-
causing mutations. The dimer state of the protein is confirmed by the structure. What is quite exciting
is the ability to convert NBCe1 from a co-transporter into an exchanger by mutating a few residues at
the substrate-binding site into those found in SLC4A1. The paper is clearly written and easy to read,
and is a significant advance to the structural biology of SLC membrane transport proteins.

We thank Reviewer 2 for providing us with insightful comments and suggestions. We are glad that the
Reviewer is excited about our NBCe1 structure-functional data and finds our manuscript a significant
advancement in SLC protein structural biology. We have addressed the Reviewer's comments in our
revised manuscript.

Minor suggestions:

1. I would include ...Human SLC4A4 Sodium-coupled... in the title.

SLC4A4 has now been added to the title as requested.

2. SLC4 and SLC26 belong to a novel class of 14TM segment inverted 7+7 fold of membrane proteins
highlighted in a 2017 review by Chang and Geertsma, which could be cited. Similarly, a definitive
2016 review of human SLC4A1 (Band 3) by Reithmeier et al., which includes a comparison with
SLC4A4 could be cited.

We have expanded our Discussion section to include a comparison between the other reported
structures and that some of the resolved structures that belong to the class of 14 TM segment with
7+7 TM inverted repeats. Furthermore, we have cited the reviews from Chang and Geertsma (ref. 31)
and Reithmeier et. al. (ref. 7) as requested.

3. DIDS may block the ion accessibility site but may also block the conformational change associated
with transport as this inhibitor is located at the interface between the gate and core domains.

This additional possibility is now included in the revised manuscript as suggested.

4. There is lots of evidence for a dimeric structure in human SLC4A1.

The discussion section has been revised as suggested.

5. The density (Fig S2) looks sufficiently detailed to identify bulky side chains and thus put the
sequence into register in the map. However, most of the side chains do not appear to be resolved.
Thus, the model can identify other side chains but perhaps shouldn't show their orientations as that
information is not experimentally determined (i.e. side chains should be labelled with their identity but
modelled as polyalanine in the PDB file). At a minimum, this limitation should be clearly stated.

We agree with the reviewer that some of the side chains do not appear resolved in our structure
based on the density representation in Supplementary Fig. 2. However, some of the EM densities
have features resembling parts of the side chain residues that can be observed at a higher map
contours. This suggests that some of these side chain EM densities are weaker compared to the
bulky side chain residues. We modeled these residues based on some of these features and
performed PHENIX RSR. It is likely difficult to observe some of these densities in Supplementary Fig.
2. These features can be better observed once the map and models are released. Additionally, we
have built a polyalanine chain for the extracellular loop 3 (EL3), as we cannot unambiguously place
the side chains in that density; even at higher map contours. We have stated it in methods of the
revised manuscript.

6. One aspect of the methodology is worrying: why was the FSC estimated with a soft spherical mask
and not the standard close-fitting mask default of Relion with correction of the FSC for the mask. It is
essential to show that the mask (even though described as loose fitting and soft edged) does not cut
into the protein density and thereby inflate the resolution estimate by FSC.

We appreciate that the Reviewer brought up a question about the mask that was used during FSC
calculation. We used the standard (close-fitting) auto-mask to calculate the FSC. A soft-edge
spherical mask (180 Å in diameter) was used during auto-refinement and not in the final FSC
calculation. We have modified the Methods section to avoid the ambiguity on which mask was used.

7. It would be useful to relate the effect of the individual cysteine mutations on transport to the
conversion to an anion exchanger with the triple mutant. For example, some single mutations like
D754E inhibit co-transport but do not yet demonstrate anion exchange.

We agree with the Reviewer regarding the usefulness of providing this additional information if
possible, however at this juncture, it would be too conjectural and beyond the resolution of the
structure to address this question satisfactorily at the single residue level.

8. It might also be useful to briefly contrast theNBCe1 structure with SLC5 sodium-dependent sugar
transporters like vSGLT.

We have added a brief comparison between these transporters in the Discussion section as
requested.

9. The section at the end of the Discussion on the advances in single particle cryo EM of membrane
proteins could be abbreviated or eliminated all-together. The use of amphipol PMAL-C8 is however,
novel and interesting (closest previously has been amphipol A8-35). This may be a better place to
more fully discuss the nature of the sodium and bicarbonate (anion) binding sites and the effect of
disease-causing mutations on protein structure.

We have abbreviated the Discussion section regarding advances in single particle cryo EM of
membrane proteins as requested. We note that amphipol PMAL-C8 has been used previously by
Paulsen et al. (Nature, 520: 511-517, 2015). Finally, we prefer at this juncture not to add further points
regarding the ion binding sites and possible effect of mutations on the structure.

Reviewer #3 (Remarks to the Author):

NCOMMS-17-20257

CryoEM Structure of the Human Sodium-Coupled Acid-Base Transporter NBCe1

Kevin W. Huynh^{1,2*}, Jiansen Jiang^{2,3*}, Natalia Abuladze^{1*}, Kirill Tsirulnikov^{1*}, Liyo Kao¹, Xuesi

Shao⁴, Debra Newman¹, Rustam Azimov¹, Alexander Pushkin¹, Z. Hong Zhou^{2,3}, and Ira Kurtz^{1,5}

Overall

This is a really wonderful structural biology study elucidating the membrane structure of human

NBCe1 at 3.9Å by cryoEM. The authors have also made some very intriguing observations

concerning a Na⁺ binding site and the anion (Cl⁻) exchange site for other SLC4 proteins (e.g., AE1).

The interdigitated TM-segments while expected from the AE1 crystal structure seem better resolved

by this solution rather than crystallization method. The weakest part of this entire manuscript is

currently the ion transport data. This does not appear to be due to inappropriateness of the

experiments but rather in the authors choices of what data to include as part of the manuscript (see

detailed comments). Overall this is a very complete study, but the authors would be well served by

highlighting specific experiments.

We thank Reviewer 3 for providing us with insightful comments and suggestions. We are glad that the

Reviewer feels that our paper is a wonderful structural biology study elucidating the membrane

structure of human NBCe1. We have addressed the Reviewer's comments in our revised manuscript.

Major Points

1) Abstract / Introduction

a) Slc4 family is typically referred to as the "HCO₃ transport" family. Why do the authors refer to this

NBCe1 protein as a Na⁺ coupled CO₃²⁻ symporter? This does not even seem to match the acronym.

The text has been revised as suggested by the Reviewer.

b) Including Slc9 as a Na⁺ coupled acid-base transport seems odd as other Na⁺ coupled-base

transport systems are not included, e.g., Slc5 monocarboxylate transporters, Slc10 Na⁺ bile acid

transporters, Slc13 Na⁺ coupled sulfate / carboxylate transporters, Slc23 Na⁺ couple ascorbic acid

transporters, Slc34 Na⁺ coupled phosphate transporters, etc

We now refer to these SLC transporter families as requested.

2) Na⁺ coupling

a) The authors have an attractive speculation for the molecular determinants of Na⁺ coupled vs. H⁺

residues (li 115-117).

We agree with the Reviewer that it is an interesting hypothesis and thank the Reviewer for the

comment.

b) The evolutionary reference seems misplaced as the authors are presuming that the Slc4 anion

exchangers are the most evolutionarily distant and the Na⁺ coupling of some Slc4-transporters is

more recent. To my knowledge there is no evidence for assigning what the Slc4 proto-transporter may

have transported. So it could be equally likely that the Na⁺ coupling "degenerated" or changed to

protonation.

We agree that either of these scenarios is possible and accordingly have removed the term

"evolutionary" from the Discussion section.

3) Figure 4 – transport data

a) Base-flux. Representative traces for 4b & 4c should be shown in Supplemental data so that readers
see the primary data and know how the authors analyze the BCECF data. Please also indicate how
the base-flux data (as well as Na⁺ and Cl⁻ dependent components) are quantified (e.g., linear or non-
linear curve fit).

Representative traces are now depicted in Supplementary Fig. 5 as suggested. How the base flux
was quantified is now specified with additional detail in the Methods section as requested.

b) Na⁺ dependent currents (4h,i, j) are very small and have different reversal potentials (compare 4i
to 4f). Why do 4f and 4i (solid symbols) differ by 4-fold. This should be the same experimental
condition, i.e., full Na⁺ with 25 mM HCO₃⁻. Perhaps since there seems to be wide variation between
preparations, the authors should show individual experiments (in the presence and absence of HCO₃⁻
189) ± Na⁺ and ± Cl⁻. The authors point that NBCe1 is a Na⁺ coupled HCO₃⁻ (or CO₃²⁻) symporter is
190 difficult to understand without clearer experimental records.

The experimental conditions used in Figs 4f and 4i differed in the original manuscript. In Fig. 4f [and
Figs 4e (mock) and 4g (mutant)], there was no bicarbonate in the pipet solution; therefore, there was
a gradient for both Na⁺ and bicarbonate when the external solution was changed from HEPES to a 25
mM bicarbonate-buffered solution. In the original experiments depicted in Fig. 4i [and Figs 4h (mock)
and 4j (mutant)], there was 25 mM bicarbonate in both the external and pipet solutions (no
bicarbonate gradient) when the external Na⁺ concentration was increased. However, because of the
Reviewer's question and the potential lack of clarity in depicting this data, we have revised the
experiments in Fig. 4i [and Figs 4h (mock) and 4j (mutant)] using a pipet solution that is now identical
to the experiments in Fig.4f [and Figs 4e (mock) and 4g (mutant)]; i.e. HEPES buffer without
bicarbonate. Increasing the external Na⁺ concentration from zero to 145 mM is now done in the
presence the same bicarbonate gradient as originally used in Fig. 4f [and Figs 4e (mock) and 4g
(mutant)] and the final experimental conditions in Fig. 4i and Fig. 4f are also now identical.
Accordingly, the Na⁺-dependent current in the experimental protocol depicted in revised Fig. 4i is now
comparable to Fig. 4f. The experiments now depicted in Figs 4h and 4j were done with the same
revised protocol. Finally, for Fig. 4f, additional experiments were repeated that confirmed the previous
results and are now averaged with the previous data.

c) Cl⁻ dependent currents (4k, l, m). presumably as in “Na⁺ dependence,” these are experiments in
the presence of 25 mM HCO₃⁻. This is not specified. If k,l,m are not HCO₃⁻ solutions, we do not learn
anything from the experiments.

The data was obtained in the presence of 25 mM HCO₃⁻. The bicarbonate-containing solutions are
more clearly stated in the revised manuscript.

212 d) Surface protein expression. Figure 4 mutational analysis relies on mutants A, B, C and A+B+C all
213 trafficking to the plasma membrane to the same extent as wt-NBCe1. Here there is not data to
214 indicate that this is true. As the authors have changed the NBCe1 amino acids to AE1 amino acids, it
is implied that there is no difference in cell surface expression. Without knowing this plasma
membrane expression and the comparative amount of expression, one cannot interpret most of the
data plotted in figures 4c-m. An additional supplement showing amount of surface expression of
A+B+C v wt-NBCe1 is minimally needed.

We now include in the Method section that the constructs were initially assessed for plasma
membrane expression using sulfo-NHS-SS-biotin plasma membrane labeling and
immunocytochemistry using a previously characterized rabbit polyclonal NBCe1 antibody as

described. As suggested, the data showing the surface protein expression of the A+B+C construct
versus wt-NBCe1 is now included in Supplementary Fig. 5.

4) Discussion

a) (li 145-147) While the investigators have a unique structure and a novel method applied to SLC4
proteins, claims that this is the “first” report of a SLC4 family member as a dimer are not true. The
early crystal structure of AE1 [Wang DN, Kuhlbrandt W, Sarabia VE, and Reithmeier RA. Two-
dimensional structure of the membrane domain of human band 3, the anion transport protein of the
erythrocyte membrane. *Embo J* 12: 2233-2239, 1993] showed AE1 as a dimer as well as a dimer of
dimers. In 2015, Arakawa and Iwata provided a 3.5 Å structure of AE1 also illustrating a dimer
[Arakawa T, Kobayashi-Yurugi T, Alguet Y, Iwanari H, Hatae H, Iwata M, Abe Y, Hino T, Ikeda-Suno
C, Kuma H, Kang D, Murata T, Hamakubo T, Cameron AD, Kobayashi T, Hamasaki N, and Iwata S.
Crystal structure of the anion exchanger domain of human erythrocyte band 3. *Science* 350: 680-684,
2015].

The Discussion section has been modified as suggested.

b) (li 162 - 164) The authors have made an important contribution, but they have overstated their
result by indicating they are modeling a 130 kD NBCe1 protein. In previous parts of the manuscript,
the authors are careful to indicate that just the membrane and near membrane regions are modeled (li
67-70). This include the cytoplasmic N-terminal Met1-Gly400 (~30-40 kD) and the cytoplasmic C-
terminus Asp966-Cys1035 (~8-9kD). In fact, lines 69-70 explicitly states that “...due to the dynamic
nature of these regions and were not modeled...” Please state what was actually reported. It seems
that ~80 kD of the membrane core of the protein was modeled. There are certainly ways to phrase the
comment that this is a big protein without claiming to model the entire protein.

The Discussion section has been modified as suggested.

c) Perhaps the authors could better discuss the merits of cryoEM structure(s) of membrane proteins v
traditional xray crystallography. To me, the cryoEM structure of a protein in solution is highly preferred
to a static crystal.

The comparison of cryoEM and X-ray crystallography in the determination of membrane protein
structure has been analyzed in recent reviews (e.g. Fernandez-Leiro & Scheres, *Nature* 537: 339-346,
2016; Wang & Wang. *Protein Sci.* 26: 32-39, 2017; Venien-Bryan et al., *Acta Crystallogr. F Struct.
Biol. Commun.* 73 :174-183, 2017). Accordingly, we feel such an analysis would be beyond the scope
of the paper.

5) Methods

a) (li 283-284) Please indicate if wt-NBCe1-A mutations were sequence verified.

The revised manuscript now states that the all constructs were sequence verified.

Minor points

• (li 21) Define “APC protein superfamily”

This has been defined as requested in the revised manuscript.

• Might be worth citing some of the actual NBCe1 literature other than just the authors published
works.

Additional references were added as suggested.

Reviewers' comments:

Reviewer #3 (Remarks to the Author):

REVISION- NCOMMS-17-20257

CryoEM Structure of the Human Sodium-Coupled Acid-Base Transporter NBCe1

Kevin W. Huynh^{1,2*}, Jiansen Jiang^{2,3*}, Natalia Abuladze^{1*}, Kirill Tsurulnikov^{1*}, Liyo Kao¹, Xuesi Shao⁴, Debra Newman¹, Rustam Azimov¹, Alexander Pushkin¹, Z. Hong Zhou^{2,3}, and Ira Kurtz^{1,5}

General.

The authors have addressed the previous comments, and I believe this is now a much better manuscript. The structural movies are superb! There are a couple of remaining issue which still require some attention.

1) the authors are being a bit obtuse with both their functional characterizations. In particular, the Na⁺ driven base flux (Figure 2) and Cl dependent base flux (Supplemental Figure 5). Since one of the extremely novel aspects of this paper is the conversion of an electrogenic Na⁺ / HCO₃⁻ (CO₃⁼) to a electroneutral Cl⁻ transporter, Supplemental Figure 5 should be one of the main figures in the manuscript. Additionally, for at least the d,e,f experiments, there must be a similar experiment with the wt-NBCe1. The authors must show that in their system and their hands, that wt-NBCe1 does not look different than their "Mock" control. This is a sticking point because there is a significant effect in the "d-trace" for the addition of Cl. Better yet, would be to show overt Cl-transport rather than pH change with Cl-addition.

2) The manuscript would be more generally accessible if the authors had a transmembrane model with explicit amino acids and numbers on the bounds of the TM segments. For NBCe1 as well as AE1 (SLC4A1), the solved membrane structures illustrate that the TM's of an individual monomer are interdigitated. For this reason in particular, more explicit label of a model in a figure are needed (e.g., table form for regions of membrane spans). I understand that this will be accessible through the mandatory sharing of the structural data; however, the MS would be more generally accessible if this analysis were merely listed at least in the Supplemental information.

3) Role of dimeric NBCe1. (line 164-165). The authors state:

"Given the dimeric state of NBCe1, it is of interest that individual NBCe1 monomers function independently within each oligomer¹⁵."

While this is the find in ref 15 (Kao et al.) it is unclear that other investigators in the field would agree. For example, Myers and Parker recently reported a human with compound, heterozygosity [Myers EJ, Yuan L, Felmler MA, Lin YY, Jiang Y, Pei Y, Wang O, Li M, Xing XP, Marshall A, Xia WB, and Parker MD. A novel mutant Na⁺ /HCO₃⁻ cotransporter NBCe1 in a case of compound-heterozygous inheritance of proximal renal tubular acidosis. The Journal of physiology 594: 6267-6286, 2016; PMID 27338124]. If lines 164-5 are true, then there should never exist a patient with compound-heterozygous as the current authors claim the monomers act independently. Either the statement should be qualified or deleted.

Minor.

- Is "mock" transfected an empty plasmid?

Summary of responses and changes in the revised manuscript/figures

We thank Reviewer 3 for the new additional comments, which we have fully addressed in this revised
manuscript.

Our responses to the reviewer's suggestions are in blue.

Reviewer #3 (Remarks to the Author):

1) The reviewer requested that the data in the original Supplementary Figure 5 be one of the main
figures in the manuscript. In addition, for the d, e, f experiments in original Supplemental Figure 5, the
reviewer requested a similar experiment with wt-NBCe1 showing that wt-NBCe1 does not look
different than the mock control.

As suggested by the reviewer, the data in the original Supplementary Figure 5 is now included in
Figure 4 in the manuscript. The electrophysiology data from original Figure 4 is now in Figure 5.

In addition, we have now included in Figure 4 an experiment with wt-NBCe1 demonstrating that it
does not differ from the mock control as requested by the reviewer.

2) The reviewer requested a new Supplementary Figure illustrating the amino acid boundaries of the
transmembrane regions in NBCe1 in comparison to AE1.

As requested, we have made a new Supplementary Figure 3 showing the alignment of NBCe1 and
AE1 with defined boundaries of each helix. We have renumbered the original Supplementary Figures
3 and 4 accordingly.

3) The reviewer requested that regarding role of dimeric NBCe1, we either remove or qualify the
sentence: "Given the dimeric state of NBCe1, it is of interest that individual NBCe1 monomers
function independently within each oligomer¹⁵."

As suggested by the reviewer, we have removed this sentence from the revised manuscript.

Minor

The reviewer questioned whether "mock" transfected is an empty plasmid.

Yes, "mock" transfected refers to an empty plasmid, which is now indicated in the Methods section.